# Glioblastoma-Specific Strategies of Vascularization: Implications in Anti-Angiogenic Therapy Resistance

**DOI:** 10.3390/jpm12101625

**Published:** 2022-10-01

**Authors:** Mariachiara Buccarelli, Giorgia Castellani, Lucia Ricci-Vitiani

**Affiliations:** 1Department of Oncology and Molecular Medicine, Istituto Superiore di Sanità, Viale Regina Elena, 299, 00161 Rome, Italy; 2Department of Neurosurgery, Fondazione Policlinico Universitario A. Gemelli IRCCS, Università Cattolica del S. Cuore, Largo A. Gemelli, 8, 00168 Rome, Italy

**Keywords:** glioblastoma, anti-angiogenic therapy resistance, glioblastoma stem-like cells

## Abstract

Angiogenesis has long been implicated as a crucial process in GBM growth and progression. GBM can adopt several strategies to build up its abundant and aberrant vasculature. Targeting GBM angiogenesis has gained more and more attention in anti-cancer therapy, and many strategies have been developed to interfere with this hallmark. However, recent findings reveal that the effects of anti-angiogenic treatments are temporally limited and that tumors become refractory to therapy and more aggressive. In this review, we summarize the GBM-associated neovascularization processes and their implication in drug resistance mechanisms underlying the transient efficacy of current anti-angiogenic therapies. Moreover, we describe potential strategies and perspectives to overcome the mechanisms adopted by GBM to develop resistance to anti-angiogenic therapy as new potential therapeutic approaches.

## 1. Introduction and Overview

Glioblastoma (GBM) is the most common and lethal primary brain tumor in adults, with a high rate of recurrence and mortality. The standard multimodal therapy for GBM, including aggressive surgery, radiotherapy, and chemotherapy, has remained unchanged for more than two decades [1,2,3]. Meanwhile, many GBM patients receiving the standard of care treatment experience a short progression-free survival (PFS), and due to the lack of effective therapies for recurrent disease, the clinical outcome is very poor. GBM is among the most vascularized of all solid tumors, and vascular proliferation is a pathological hallmark of GBM [4]. GBM vasculature is typically composed of abnormal glomeruloid vascular structures consisting of complex aggregates of newly formed microchannels, lined with hyperplastic endothelial cells characterized by an altered morphological phenotype and supported by basal lamina and pericytes [5]. These vessels are structurally and functionally abnormal, contributing to a hostile microenvironment, and their presence is the key histopathological characteristic that discriminates GBM from lower-grade gliomas contributing to the generation of a more malignant phenotype with increased morbidity and mortality [5]. Hence, the introduction of the recombinant humanized monoclonal antibody anti-VEGF (vascular endothelial growth factor), bevacizumab, approved in 2009 in the United States as an anti-angiogenic adjuvant therapeutic strategy for recurrent GBM, raised initial enthusiasm among clinicians, but unfortunately, it was rapidly attenuated due to its limited survival advantage compared with experimental therapy at recurrence [6]. Indeed, despite notable radiographic responses and improved PFS, the effects of anti-angiogenic therapy are unfortunately not long-lasting. Currently, several alternative cellular and molecular mechanisms of vessel recruitment and tumor angiogenesis have been described in GBM that contribute to tumors escaping from anti-angiogenic therapy and even evolving into a more aggressive phenotype.

Traditionally, new vessel formation, within the tumor mass, has been ascribed to the sole process of neoangiogenesis and simply described as capillary sprouting from pre-existing capillaries. Through this process, the growing solid tumors generate an increased blood supply to satisfy their increasing nutrient and oxygen demand. It is now recognized that tumor angiogenesis is a multi-step, finely tuned process in which a highly complex spectrum of events occurs.

Along with the most studied mechanisms of neovascularization, i.e., angiogenesis and vasculogenesis, at least three distinct mechanisms of neovascularization in GBMs have been identified: (i) vessel co-option, (ii) vasculogenic mimicry (VM), and (iii) GBM stem-like cell (GSC) transdifferentiation [7]. These mechanisms are not independent but are rather closely interconnected and triggered by hypoxia as a major stimulator of angiogenesis in GBM [8,9]. Temporally, vessel co-option, the process whereby tumors utilize native brain vessels to recruit blood supply, is the first mechanism by which gliomas achieve their vasculature, followed by angiogenesis, the process by which new vessels are obtained from pre-existing ones. The weight of vasculogenesis, i.e., the differentiation of the circulating bone-marrow-derived cells, in tumor neovascularization is still controversial, and animal studies of tumor-associated vasculogenesis produced discordant conclusions [10,11]. Differently from vasculogenesis, VM, defined as the ability of tumor cells to form functional vessel-like networks, has been described first in human melanoma models [12] and later in human astrocytoma samples [13]. The most recently described mechanism of GBM neovascularization consists of the direct transdifferentiation of GSCs [14,15]. This mechanism is complementary to VM, but even though no lines of evidence exist for a direct relationship between hypoxia and VM, hypoxia is the main driver for transdifferentiation, as hypoxia and the hypoxia-inducible factors (HIFs) play critical roles in maintaining and supporting the survival and self-renewal of GSCs [16]. Despite the significant hypoxia-induced production and secretion of VEGF by GSCs, the transdifferentiation process is VEGF-independent.

Here, we review the GBM-associated mechanisms of neovascularization and their implication in treatment resistance that characterizes GBM. In addition, we review the rationale for the targeting of neovascularization in gliomas (e.g., anti-angiogenic therapeutic strategies) and discuss the potential molecular mechanisms that could explain escape from anti-angiogenic therapy. Finally, we describe potential strategies and perspectives that should help overcome mechanisms of resistance to therapy paving the way toward new potential therapeutic approaches.

## 2. GBM-Associated Mechanisms of Neovascularization

### 2.1. Vessel Co-Option

Vessel co-option, also known as perivascular migration, is a non-angiogenic process whereby cancer cells co-opt and utilize the pre-existing vessels of the surrounding normal tissue for their growth and spread. Vessel co-option is independent of the classic angiogenic switch and occurs in the absence of angiogenic growth factors [17]. To meet their metabolic demands, tumor cells migrate along the abluminal surface of pre-existing vessels and/or infiltrate the tissue space between them, leading to the incorporation of the pre-existing vasculature into the tumor [18]. Initially observed in lung cancers with alveolar non-angiogenic growth patterns [19], vessel co-option was further described in several other cancer types including gliomas, in which the tumor cells organize themselves into cuffs around normal microvessels [7]. In GBM, vessel co-option was first described by Holash et al., following the injection of rat C6 glioma cells into the rat striatum. They suggested the term “vessel co-option” to indicate this non-angiogenic phenomenon in which the co-opted vessels were characterized for the expression of angiopoietin-2 (ANG-2) [20]. The brain is a highly vascularized tissue; thus, vessel co-option represents one of the different invasion pathways used by glioma cells for the infiltration of the surrounding brain tissue [21]. Different patterns of tumor growth involving the vessel co-option mechanism have been described in the brain. In the perivascular pattern, tumor cells migrate in the perivascular space and adhere to the surface of the pre-existing vessels. In the diffusely infiltrating pattern, cancer cells infiltrate the stromal space by moving in between host vessels without creating permanent contact with them [22].

Distinctive morphological features are associated with vessel co-option and can be used to distinguish this process from angiogenesis through histological examination. The maintenance of the vascular architecture of the normal brain tissue in the infiltrated areas is one of the clearest signs of vessel co-option, in contrast to the abnormal and chaotic vasculature typical of tumor angiogenesis [18]. Moreover, vessel co-option generates histological pictures, such as satellites and finger-like extensions, depending on the cutting plane [23]. In terms of immunohistochemical markers, Baker et al. examined human GBM patient biopsies containing the infiltrative tumor margin for the presence of vimentin and the von Willebrand factor (vWF). The immunolabeling of these two markers showed perivascular-associated vimentin^+^ tumor cells at the transition zone between tumor and normal tissues [24]. However, there are no clinically validated techniques and biomarkers that can be used to undoubtedly identify vessel co-option in GBM patients and to argue its predictive and/or prognostic significance.

In gliomas, the process of tumor vascularization is spatially and temporally dynamic in terms of glioma cell–vascular interactions and the mechanisms utilized for tumor growth. Temporally, it has been shown that vascular co-option is the first mechanism by which gliomas achieve their vasculature and precedes angiogenesis by up to 4 weeks [7]. In particular, it appears to be the dominant mechanism of vascularization in early-stage rat C6 gliomas, followed by the rapid development of angiogenesis due to vessel regression and hypoxia [20]. However, in mouse GL26 and rat CNS-1 orthotopic gliomas, vessel co-option has been described as an iterative growth process that occurs throughout the entire course of brain tumor progression [24]. Likewise, different ways of glioma cell-vasculature interactions have been described using different orthotopic glioma models. By using the U87MG brain xenograft mouse model of GBM, Watkins et al. examined glioma–vascular interactions at a much earlier disease stage, showing that the invading glioma cells colonize the perivascular area of pre-existing vessels, wrap themselves around the abluminal surface, and displace astrocytic endfeet from endothelial or mural cells. This causes a local breach of the blood–brain barrier (BBB), followed by abnormal permeability and impaired functionality [25]. Recently, in a patient-derived GSC xenograft model, Pacioni et al. observed a variety of interactions between glioma cells and perivascular astrocytes, most of which do not imply the disruption of the BBB. In particular, they showed that tumor cells invading perivascular spaces are spatially associated with vessels showing preserved BBB [26].

From a molecular point of view, the mechanisms that occur in the course of glioma vessel co-option are still being investigated. Several soluble factors, membrane receptors, and adhesion molecules have been described as critical for co-opting cells to move along vessels. Among them, bradykinin, epidermal growth factor receptor variant III (EGFRvIII), C–X–C chemokine receptor type 4 (CXCR4)/stromal-cell-derived factor 1α (SDF-1α), and Wnt7/Olig2 pathways promote vessel co-option [27,28,29,30]. Adhesion molecules, such as integrins, L1 cell adhesion molecule (L1CAM), cell division cycle 42 (CDC42), and ephrin-B2 (EB2), are required for the morphological changes occurring in glioma co-opting cells, leading to the emergence of a mesenchymal-like phenotype [31,32,33,34]. Recently, the downregulation of the NOTCH1 signaling pathway has been associated with decreased vessel co-option and reduced perivascular tumor cell population [35]. It has been widely documented that vessel co-option is a VEGF-independent mechanism of tumor vascularization, mainly driven by the proangiogenic factor ANG-2 without an increase in VEGF expression, and activation of matrix metalloproteinase-2 (MMP-2), which promotes glioma cell invasion [36]. As a consequence of co-option, the engulfment of the pre-existing brain vasculature during glioma growth leads to the compression of vessels around co-opting cells, with resulting hypoxia, vascular regression, and increased expression of VEGF [37].

#### 2.1.1. Vessel Co-Option and Implications in Anti-Angiogenic Resistance

Vessel co-option has been reported in both high-grade and low-grade gliomas [32,38]; it appears to occur mainly at the tumor–brain interface and in the peritumor regions, contributing to tumor infiltration [21]. It might be assumed that vessel co-option and angiogenesis occur where the perivascular niche is set around a vessel with microvascular sprouting or vascular cluster conformation, as in the moderately aggressive GBM types or regions [39]. Vessel co-option and angiogenesis can occur simultaneously in different areas of the same tumor, or a switch between the two mechanisms can occur over time [18]. Vessel co-option has been observed in untreated brain tumors, but more and more experimental data and clinical studies propose vessel co-option as an emergent mechanism of resistance to anti-angiogenic therapy in gliomas.

In the 2000s, Rubenstein et al. and Kunkel et al. demonstrated that anti-VEGF treatment of orthotopic brain tumors in immunodeficient mice led to the increased infiltration and co-option of the host vasculature, despite a decrease in tumor vascularity and proliferation [40,41]. In the 2010s, vessel co-option was observed in human GBM patients following anti-VEGF therapy with bevacizumab [42,43], contributing to the so-called “infiltrative shift” through which the tumor spreads towards perivascular spaces. In bevacizumab-treated orthotopic mouse models, an enhancement in perivascular growth was observed, which is mediated by the triggering of a mesenchymal signature involving different signaling pathways, such as the receptor plexin domain containing 1 (PLXDC1) [23] and the tyrosine kinase c-Met [44]. Another study by Huveldt et al. reported Src family kinase hyperactivity at the tumor edge of the GBM orthotopic model in response to bevacizumab, which may contribute to increased cell motility and invasion [45]. Griveau et al. reported an increased Wnt7a expression in human GBM patient biopsies after bevacizumab treatment and a selection for the Olig2^+^/Wnt7^+^ phenotype in a mouse model following prolonged anti-VEGF therapy [30].

It has also been reported that treatment with the pan-VEGF receptor tyrosine kinase inhibitor, cediranib, leads to an increase in tumor infiltration along the pre-existing brain vessels, both in animal models [37] and human GBM patient biopsies [46].

A proinvasive evasion mechanism has also been described following the treatment with the VEGF inhibitor aflibercept in mice bearing GL261 gliomas [47]. Treatment with this receptor decoy for VEGF-A resulted in increased tumor invasiveness [47], which might be assumed to explain the lack of benefits in tumor growth inhibition observed in the U87MG GBM mouse model [48].

#### 2.1.2. Vessel Co-Option and Potential Strategies to Overcome Resistance to Anti-Angiogenic Treatment

The preference of glioma cells in the perivascular space as a route for invasion following anti-angiogenic treatments may be explained as a prosurvival response to decreased oxygen and nutrients, which in turn, activate the hypoxia-triggered induction of invasion-related pathways. For this reason, the inhibition of vessel co-option could be clinically relevant to reducing the occurrence of acquired resistance and improving the efficacy of current anti-angiogenic therapies.

Several studies performed on orthotopic brain tumor models showed increased effectiveness in the suppression of tumor growth and improvement in survival by combining anti-angiogenic drugs with blocking co-option-related mechanisms. It has been reported that the combination of PLXDC1 inhibition and bevacizumab treatment is able to prolong survival, preventing bevacizumab-induced infiltrative growth [23]. Similarly, Huveldt et al. demonstrated that treatment with the Src family kinases inhibitor dasatinib effectively counteracts the bevacizumab-induced invasion and infiltration of GBM cells in orthotopical xenografts [45]. On this point, a mathematical model developed by Voutouri et al. suggested that sequential blockade of co-option first followed by VEGF inhibition is likely to be more beneficial than simultaneous treatment [37].

However, the contribution of vessel co-option to therapy resistance can be considered as an innate/intrinsic feature of the tumor, regardless of the anti-angiogenic treatment response. Therefore, targeting this mechanism might impact the clinical efficacy of glioma therapy. In line with this hypothesis, a potentially valuable treatment approach has been described by Renner et al., showing that the combined administration of aflibercept and picornavirus-based anti-tumor vaccine, TMEV Xho1-OVA8, to mice bearing GL261 gliomas leads to a delay in tumor progression and improved survival [47]. In an orthotopic brain tumor model, Yadav et al. reported that targeting perivascular invasion by the downregulation of CXCR4 makes the tumor more sensitive to radiation therapy [29]. Griveau et al. showed that blocking Wnt signaling with the porcupine inhibitor LGK974 improves the response to temozolomide (TMZ) treatment in a proneural GBM patient-derived in vivo model, improving survival.

Over the last few years, therapeutic approaches based on the inhibition of vessel co-option alone or in combination with bevacizumab treatment have been evaluated in clinical trials. Among these, the dual inhibitor of Met and VEGFR-2 cabozantinib [49], the Akt inhibitors NVP-BKM120 and perifosine [50,51], and the hypoxia-activated prodrug evofosfamide [52] showed limited clinical impact in improving the effects of current anti-angiogenic therapy. On this point, the overall efficacy of potential novel treatments might be affected by the inability to stratify glioma patients based on the occurrence of angiogenesis or vessel co-option [21]. For this reason, a deeper understanding of the mechanisms underlying the reciprocity between these two strategies of tumor vascularization could allow a personalized and more effective therapeutic approach.

### 2.2. Vasculogenic Mimicry

Vasculogenic mimicry (VM) is a new mechanism of tumor neovascularization in which highly invasive and genetically dysregulated tumor cells acquire vascular cell features or function, forming de novo vascular-like structures. These structures mimic the function of blood vessels, thus providing an adequate blood supply for tumor growth and metastasis [53]. The functional role of VM structures in tumor circulation has been demonstrated by different approaches, such as the microinjection method, Doppler ultrasonography, magnetic resonance imaging, and laser scanning confocal angiography [54,55,56,57,58,59]. VM was firstly discovered in uveal melanoma as the formation of a circulatory system by dedifferentiating tumor cells [12]. VM structures were characterized by the abundance of matrix proteins (proteoglycans, laminin, collagen IV, and VI), negative staining for endothelial markers (CD31 and CD34), positive staining for periodic acid–Schiff (PAS), and the presence of blood components (such as erythrocytes, platelets, and hemoglobin) in their lumen [12,53,60].

Subsequently, VM was described in other solid tumors. The first evidence for VM in gliomas was reported by Yue and Chen in 2005. The authors examined 45 cases of WHO II-IV grade astrocytoma tissues by the dual staining of CD34 and PAS to investigate whether VM existed in these tumors. They reported that the tumor microvasculature was mainly composed of endothelium-lined vessels that stained positively for PAS, laminin, and endothelial markers. However, they found that 2 out of the 45 astrocytoma tissues had PAS^+^/CD34^−^ vessels containing red blood cells [13]. Liu et al. found an association between microvascular density (MVD) and VM in gliomas. VM-positive gliomas had low MVD compared with VM-negative gliomas, suggesting that VM structures could represent a complementary mechanism to sustain blood supply, particularly in areas with low MVD [61]. Furthermore, the authors observed a positive correlation between VM and the WHO grade of glioma. The patients with VM-positive gliomas had shorter overall survival than those with glioma without VM [61]. As described in other tumors, VM might be of two types in GBM: a patterned matrix type of secreting matrix proteins and a tubular type characterized by tumor cells lining the vessel-like structures [62].

In GBM, the involvement of GSCs in VM has been reported by several studies [62,63,64]. Chiao et al. demonstrated that CD133^+^ GSCs contributed to forming VM in tumor xenografts, particularly the CD133^+^ GSC-derived xenografts showed vessel-like structures negative for CD31 staining and positive for PAS and α-smooth muscle actin (α-SMA), suggesting that these cells may contribute to forming vessel-like structures by transdifferentiating in vascular smooth muscle-like cells. Likewise, other studies reported that GSCs might differentiate into vascular smooth muscle-like cells or vascular mural-like cells to induce VM of the tubular type in GBM [62,64].

The molecular mechanisms underlying VM formation involve a complex network of signaling pathways. Hypoxia seems to be one of the main inducers of this process, and through its main effector, HIF-1α, it directly regulates several VM-related effectors such as VE-cadherin (CDH5) and MMPs [65,66]. MMP-14, MMP-9, and MMP-2 induce matrix remodeling, which promotes VM in glioma [67,68]. MMPs are also regulated by upstream regulators such as the transforming growth factor-β (TGF-β), which has a critical role in VM formation (Ling et al. 2011 [69]). When activated by upstream effectors, MMP-14 activates MMP-2, inducing the cleavage of the laminin subunit-γ2 (LAMC2) chain into promigratory γ2′ and γ2, which in turn promote VM formation [67,69]. TGF-β promotes the expression of other adhesion molecules such as CDH5 [70]. It has been reported that CDH5 and erythropoietin-producing human hepatocellular receptor A2 (EphA2) are highly expressed in VM-positive glioma compared with VM-negative glioma, and their expression is required for VM formation [65,71]. CDH5 modulates the EphA2 activity, which in turn regulates p85, the regulatory subunit of phosphoinositide 3-kinase (PI3K), promoting the loss of intercellular adhesion and facilitating migration and infiltration [72,73]. It has been demonstrated that the EGFR/PI3K/AKT/mammalian target of the rapamycin (mTOR) pathway is closely related to VM formation, as well as other signaling pathways, including the CXCR4/AKT, insulin-like growth factor-binding protein 2 (IGFB2), VEGF/VEGFR-2, and interleukin 8 (IL-8)/CXCR2 pathways [74,75,76,77,78,79,80]. Notably, VEGF/VEGFR-2 can stimulate VM, inducing EphA2 to enhance MMP expression, thus favoring extracellular matrix remodeling [67,76,81]. As expected, with such a complex process, the regulatory roles of non-coding RNAs (ncRNAs), particularly miRNAs and lncRNAs, have widely been reported in the induction of VM in glioma [68,71,82,83,84,85,86,87,88,89].

#### 2.2.1. Vasculogenic Mimicry and Implications in Anti-Angiogenic Resistance

In the last few years, VM has received more attention as an emerging mechanism implicated in anti-angiogenic therapy resistance in GBM. The disruption of GBM vasculature through anti-angiogenic therapies induces a hypoxic microenvironment that promotes VM as an adaptative strategy to enable GBM to survive and progress when angiogenesis is blocked [90]. Evidence of VM as a consequence of anti-angiogenic therapies has been reported in experimental models. In orthotopic GBM xenograft models, the pan-VEGFR inhibitor vatalanib and bevacizumab induced VM [80,81,91]. Following vatalanib treatment, Angara et al. reported that treated mice developed larger tumors than control mice. Vatalanib-treated tumors were characterized by extremely hypoxic areas and more PAS^+^ vascular-like structures. A high expression of HIF-1α at the core of treated tumors suggests that hypoxia following vatalanib treatment might accelerate VM formation. A positive correlation between the increased size of the tumors and the increased incidence of VM in the hypoxic areas of tumors has been reported [91]. Bevacizumab treatment induces a reduction in tumor growth and microvascular density but an increase in VM in an intracranial transplantation GBM model [92]. Accelerated VM was observed in the core and at the periphery of bevacizumab-treated tumors [80]. In orthotopic U87MG GBM mouse models, an increase in VM was observed 6 days after bevacizumab treatment, while the amount of the other neovascularization mechanisms (i.e., sprouting angiogenesis, vascular co-option, and intussusceptive microvascular growth) decreased, suggesting that VM may precede tumor cell invasion and vascular co-option in response to anti-angiogenic therapy [93].

#### 2.2.2. Vasculogenic Mimicry and Potential Strategies to Overcome Resistance to Anti-Angiogenic Treatment

Since VM participates in the anti-angiogenic resistance of GBM, it represents a promising target for developing innovative and alternative therapeutic strategies. Recently, several studies have focused on identifying the strategies able to target VM-related proteins and inhibiting this process.

Interfering with the VM-related proteins associated with the signaling pathways described above can inhibit VM formation in xenograft tumor models. SU1498 and AZD2171, which are VEGFR-2 kinase inhibitors, have been demonstrated to inhibit VM in glioma cell lines, concomitant with a reduction in tumor proliferation and tumorigenicity [94]. Likewise, galunisertib (LY2157299), a selective ATP-mimetic inhibitor of TGF-βRI, is currently under clinical trials in glioma patients and affects VM formation. This compound exerts an inhibitory effect on VM through the downregulation of CDH5, α-SMA, and Akt and VEGFR-2 phosphorylation in glioma cell lines and orthotopic A172 GBM mouse models [70]. The Akt/mTOR pathway is closely associated with VM formation in GBM; indeed, in vitro studies demonstrated that the disruption of this pathway can inhibit VM formation in glioma cells [77,95]. Recently, Zhu et al. demonstrated that celastrol, a triterpenoid derived from Chinese herbal medicine, disrupted VM formation by blocking the PI3K/Akt/mTOR pathway in orthotopic GBM xenografts. Additionally, following celastrol treatment, the authors reported a reduction in the expression levels of VM-related proteins such as CDH5 and EphA2 [96]. A reduction in VM structures, both at the core and at the periphery of tumors derived from orthotopic GBM xenografts, has been observed following treatment with HET0016, a selective inhibitor of 20-HETE synthesis by regulating the enzymes of the cytochrome P450 families [91]. Histone deacetylase (HDAC) inhibitors have also been identified as potentially targeting VM [97]. HDAC3 expression is associated with the presence of VM in glioma tissues and contributes to VM in gliomas, possibly through the PI3K/ERK and MMPs/LAMC2 pathways [67]. The effect of vorinostat, trichostatin A (inhibitors of class I and II HDACs), entinostatas (inhibitor of class I and III HDACs) and MC1568 (inhibitor of HDAC class II) has been evaluated on VM formation in GBM. This in vitro study demonstrated that HDAC inhibitors are able to impair glioma cell line and GSC capability to form tube-like structures on extracellular matrix, as a model of VM [97]. Recently, N6-isopentenyladenosine (iPA), an adenosine modified with an isopentenyl chain, a product of the mevalonate pathway, has been identified as a compound able to interfere with VM. IPA impaired this process by affecting the cytoskeletal structure of GBM cells reducing the formation of VM structures on matrix in vitro by the modulation of the Src/p120-catenin pathway and the inhibition of RhoA-GTPase activity [98].

The combination of anti-angiogenic therapies and the agents able to interfere with VM-related proteins could represent a promising therapeutic approach. SB225002, a CXCR2 inhibitor, significantly disrupted the tube-forming capability of glioma cell lines in normoxic and hypoxic conditions. Moreover, the combination of SB225002 with the anti-angiogenic drug, vatalanib, dramatically impaired VM, reducing the PAS^+^/laminin^+^ VM structures in orthotopic GBM models [80].

Nanoparticle-based strategies have been developed to interfere with VM formation in GBM, such as NGR-modified liposomes containing combretastatin A4 (NGR-SSL-CA4) and liposomes incorporating drugs (epirubicin and celecoxib) [99,100]. Notably, epirubicin/celecoxib liposomes are able to cross the BBB, accumulating in the tumor areas of the brain and destroying VM structures in orthotopic xenograft mouse models [100]. Oligonucleotide-based therapies are emerging as innovative and alternative therapeutic approaches in GBM [101]. So far, several miRNAs (miR-29a-3p, miR-584-3p, let-7f, miR-9, and miR-26b) have been described as important regulators of VM formation; thus, they may be interesting candidates as therapeutic agents in the form of miRNA mimic [71,82,83,84,89]. However, these studies are still preliminary, and further studies are needed to understand the complex networks of interactions coordinated by miRNAs in VM, to minimize off-targets effects, paving the way to a more effective targeted delivery system for possible clinical applications.

### 2.3. Cell Transdifferentiation

Cell transdifferentiation is a process whereby GBM cells have the ability to acquire an endothelial and/or pericyte phenotype, contributing to the formation of the tumor vasculature. The hypothesis of the endothelial transdifferentiation of tumor cells was first described in human cutaneous melanoma models [102,103]. Then, it was demonstrated that primary GBM cell lines and GSCs can be induced to differentiate in the cell types of the mesenchymal lineage [104,105]. Similar to normal neural stem cells (NSCs), able to generate glial and neuronal lineages, pluripotency is a typical feature of GSCs. The GBM–endothelial cell transdifferentiation represents one of the most recent GBM-associated neovascularization mechanisms that has been described. In 2010, two groups independently demonstrated the transdifferentiation of GSCs into ECs in vitro and the role of GSCs in tumor endothelium [14,15]. Specifically, Ricci-Vitiani et al. demonstrated that a proportion of CD31^+^ endothelial cells shared the same chromosomal alterations as the tumor cells within GBM specimens. Moreover, they showed that a significant fraction of GFAP^+^ microvascular cells displayed an aberrant glial/endothelial phenotype. Interestingly, in mouse GSC xenografts, about 70% of the CD31^+^ cells from the inner part of the tumor were of human origin [14]. Wang et al. reported that the fraction of CD105^+^ endothelial cells harboring the amplification of *EGFR* and the centromeric portion of chromosome 7 was similar to that of the tumor cells themselves. Furthermore, following the experiments with dissociated human GBM specimens, the authors postulated that the CD144^+^/CD133^+^ double-positive population represents the endothelial progenitor cells (EPCs) that arise from the CD133^+^ population and can differentiate into an endothelial phenotype [15]. Under specific culture conditions that promote endothelial differentiation, GSC-derived endothelial cells (GdECs) show a typical flagstone vascular endothelial cell morphology and the ability to form tubular-like structures when cultured in Matrigel [106,107]. GdECs express specific vascular endothelial cell markers, such as CD31, Tie-2, VEGFR-2, vWF, but also EPC-specific markers, such as CD34 [14,15,108]. Interestingly, an analysis of surgical GBM specimens showed tumor vessels co-expressing markers of early vascular endothelial cells (CD34) and GSCs (ABCG2 and nestin), suggesting the existence of interim cells during the transdifferentiation process [107]. This hypothesis might be supported by the findings that, under proangiogenic conditions, the glioma cells incorporated into the tumor vasculature lost GFAP expression and gained CD133 expression, shifting to a more stem/progenitor phenotype [109]. Intriguingly, a recent study reported that the tumor xenografts originating from CD34^high^-expressing GdECs showed a more undifferentiated phenotype compared with CD34^-/low^ expressing cells [108].

It has been demonstrated that GBM cells are also able to transdifferentiate into pericyte-like cells [110]. Cheng et al. reported that GSCs have the capacity to acquire a pericyte lineage phenotype in vitro, characterized by the expression of typical pericyte markers such as CD146, α-SMA, neuron–glial antigen 2 (NG2), and CD248. Analyzing human GBM specimens, they also showed that the vast majority of pericytes were of neoplastic origin, carrying the same genetic alterations as cancer cells [110].

The tumor microenvironment plays a remarkable role in the transdifferentiation process. The interaction between tumor cells and the endothelium is bidirectional, and their roles seem to be interchangeable, depending on the microenvironment demands [39,111]. A study on human GBM tissues reported the existence of a perivascular niche in which GSCs are close to CD34^+^ endothelial cells [112]. This has been described as a favorable environment that allows for the crosstalk between tumor cells and endothelial cells, thus promoting GSC maintenance and vascular development [113]. A recent study by Zheng at al simulated a perivascular niche through a hypoxic co-culture system in vitro, showing that GSCs transdifferentiate in nestin^+^/CD31^+^ cells, whose frequency in the histological samples of GBM correlated with a poor prognosis [114]. The distribution of tumor-derived ECs does not appear to be homogeneous within the tumor, but these cells were more abundant in the core rather than in the tumor periphery [111]. Interestingly, this distribution correlated with the high density of GSCs found in the hypoxic core of the tumor [115]. Indeed, it has been proven that hypoxia represents the main driver for GSC transdifferentiation [106,116], controlling stem cell self-renewal and plasticity, promoting the formation of pseudocapillary structures laid on a matrix structure and GdECs [117]. Moreover, endothelial transdifferentiation has been described as a VEGF-independent mechanism [111]. Several studies reported the involvement of the NOTCH signaling pathway in this process [114,118,119]. In particular, Zheng et al. reported that nestin^+^/CD31^+^ cells of the hypoxic perivascular niche showed a high expression of NOTCH–ligands JAG1 and DLL4, also suggesting the role of these factors in mediating the interaction with GSCs [114]. Furthermore, Hu et al. demonstrated that GSC transdifferentiation into GdECs is mediated by the epigenetic activation of Wnt5a, through Akt signaling, and promotes host EC recruitment to create a vascular niche sustaining GSC growth and survival [120]. Recently, it has been reported that the P4HA1/COL6A1 signal axis can drive the transdifferentiation of GSCs into GdECs, promoting the expression of the endothelial marker CD31, thus contributing to the neovascularization process in response to the hypoxic microenvironment [121]. The authors also showed the co-expression of P4HA1 and CD31 in endothelial cells within blood vessels in human glioma specimens, other than a positive correlation between P4HA1 and the blood vessel density [121].

The transdifferentiation of GSCs into pericyte-like cells has been described to rely on the recruitment of GSCs by tumor endothelial cells via SDF-1/CXCR4 signaling and the generation of pericytes through the activation of the TGF-β pathway [110]. It has been also reported that GSC-derived pericytes depend on VEGFR-2 expression [64] or on the activation of the NOTCH signaling pathway [122]. Recently, it has been suggested that EGFR and NF-kB signaling are involved in GSC transdifferentiation to pericytes [123]. Two distinct vascular phenotypes associated with different statuses of the *EGFR* gene have been described, characterized by GSC-derived pericytes closely related to ECs or delocalized with the disruption of the BBB [123,124]. Interestingly, an emerging regulatory mechanism of this transdifferentiation process involves netrin-1, a protein recently postulated as a non-canonical angiogenic ligand [125]. In particular, it has been hypothesized that the netrin-1 contained in exosomes secreted in the microenvironment could interact with the receptor UNC5, trigger the activation of NF-kB, and regulate the transdifferentiation to pericyte [125].

#### 2.3.1. Cell Transdifferentiation and Implications in Anti-Angiogenic Resistance

The transdifferentiation ability of GSCs is a process largely independent of VEGF signaling and, therefore, insensitive to anti-angiogenic therapies [111]. It has also been reported that tumor-derived ECs are able to undergo an endothelial-to-mesenchymal transition (EndoMT), a process whereby endothelial cells acquire mesenchymal features [90]. Liu et al. demonstrated that the platelet-derived growth factor (PDGF) signaling pathway mediates the EndoMT process in tumor-derived ECs in GBM, mainly downregulating the VEGFR-2 expression and inducing resistance to anti-angiogenic treatment [126]. The inhibition of the VEGF/VEGFR signaling pathway, using anti-angiogenic therapies, can trigger intratumoral hypoxia, which is one of the main inducers of the GSC transdifferentiation process [106,116]. Hypoxia induces an increase in the GBM cell population expressing CD133 within the tumor and promotes the self-renewal and proliferation of CD133^+^ cells in cultures derived from GBM tissues by inducing a set of key genes, including HIF-1α, HIF-2α, etc. [127,128,129,130]. In an orthotopic GBM mouse model, Soda et al. found that tumor-derived ECs were mostly present in the hypoxic, deep areas of the GBM rather than in the periphery, suggesting hypoxia as a critical stimulus for this process. Furthermore, anti-angiogenic therapies lead to increased tumor-derived ECs, supporting the hypothesis that these cells contribute to the anti-angiogenic therapy resistance of GBM [111]. Notably, bevacizumab treatment blocks tumor endothelial progenitor differentiation to mature ECs, but it does not inhibit CD133^+^ cell differentiation into endothelial precursors [15]. In xenograft models, it has been reported that TMZ, combined or not with bevacizumab, promotes tumor-derived EC incorporation into blood vessels [36].

Moreover, chemotherapy and radiation therapy might increase the GSC subpopulation and tumor-derived ECs. Ionizing radiation could enhance GSC transdifferentiation into tumor-derived ECs. Irradiated GSCs express Tie2, increase VEGF-dependent migration, form more pseudotubes in Matrigel in vitro, and develop more functional blood vessels in Matrigel plugs implanted in nude mice [131]. Following radiation therapy, in recurrent GBM, the fraction of cancer cells that express endothelial markers increases up to 2–2.7 folds, compared with GBM at primary surgery. In this context, GSCs play a relevant role in the neovascularization after the radiation-induced senescence of the brain endothelium [132]. Thus, GSC transdifferentiation is implicated in anti-angiogenic therapy resistance as well as in the revascularization following chemotherapy and radiation therapy. The observation that GdECs are incorporated into blood vessels in human GBM specimens has been reported by different studies [14,15,120]. In particular, Hu et al. found a higher level of Wnt5a and GdEC expression in peritumoral and recurrent GBMs compared with the matched intratumoral and primary tumors, respectively, suggesting that GdECs support tumor growth and invasion, which contribute to recurrence. Thus, the authors hypothesized that targeting this process could represent a strategy to limit neovascularization in human GBM patients undergoing VEGF therapy [120].

In 3D mathematical modeling, anti-angiogenic and anti-mitotic therapies induce a reduction in tumor size but enhance invasiveness. Anti-GSC therapies, which alter the stem cell niche or induce differentiation, reduce GBM invasiveness but are ultimately limited in reducing tumor size because GdECs maintain GSCs by releasing the crosstalk factors that sustain their self-renewal. Thus, the authors suggested that a combination of regimens targeting both GSCs and GdECs could reduce GBM growth and invasion and could eradicate GBM without recurrence [133]. For this reason, a deeper understanding of GSC transdifferentiation as a mechanism of neovascularization is fundamental to unveiling the GdEC contribution to anti-angiogenic therapy resistance and identifying alternative therapeutic approaches to combine with anti-angiogenic therapy in order to overcome resistance.

#### 2.3.2. Cell Transdifferentiation and Potential Strategies to Overcome Resistance to Anti-Angiogenic Treatment

A 3D mathematical model developed by Yan et al. suggested that a combination of current standard therapy (radiotherapy, TMZ, and anti-angiogenic therapies) with treatments that target GSC transdifferentiation could represent a chance to overcome therapeutic resistance and thus lead to GBM eradication [133]. Recently, several strategies targeting GSC transdifferentiation have been proposed. Although the signaling pathways involved in this process are not yet fully elucidated, the NOTCH pathway seems to have a critical role. Indeed, interfering with this signaling pathway can alter GSC transdifferentiation. The disruption of the NOTCH signaling pathway with NOTCH1 silencing or treatment with DAPT, a γ-secretase inhibitor, reduces tumor-derived ECs in GBM organotypic cultures [15,134]. DAPT treatment suppresses the transition from CD133^+^ cells to endothelial progenitor cells [15]. Likewise, the silencing of the NOTCH1 activator b1,4-galactosyltransferase V (b1,4GalTV) prevents the transdifferentiation of GSCs into ECs in an intracranial glioma model [119]. It has been also reported that NEO212, a conjugate of TMZ and perillyl alcohol, besides blocking the EndoMT process in vivo, is able to inhibit the TGF-β and NOTCH pathways, thus limiting the invasion and tubule formation of tumor-derived ECs [135].

The involvement of the P4HA1/COL6A1 axis in GSC transdifferentiation has been recently described [121,136]. Zhou et al. reported that P4HA1 knockdown promotes the expression of VEGF165b, an anti-angiogenic isoform of VEGF-A, and inhibits the intracranial tumor growth and transdifferentiation of GSCs into ECs in xenograft mouse models, suppressing GBM neovascularization. P4HA1 inhibits collagen IV synthesis and thus disrupts the structures of vascular basement membranes in GBM [136]. Thus, P4HA1 could represent a potential new target for interfering with GSC transdifferentiation.

As an alternative strategy to overcome anti-VEGF resistance, tumor-derived EC-specific targeting by the pharmacological inhibition or genetic deletion of the PDGF signaling pathway significantly sensitizes tumors to anti-VEGF therapy. In particular, the pharmacological dual inhibition of PDGFR-β and VEGFR-2 reduces GBM vascularization in GBM mouse models and improves animal survival, mainly interfering with the EndoMT process of tumor-derived ECs, which exhibit resistance to anti-VEGF/VEGFR-2 treatment [126]. Since VEGFR-2 expression in GSCs plays important role in VM and pericyte transdifferentiation [64], it might be hypothesized that the dual inhibition of PDGF and VEGFR-2 signaling pathways might act on GSC and pericyte and could represent a strategy for the complete disruption of GBM vasculature. The importance of targeting both GSC and transdifferentiation-derived subpopulations has also been revealed by the results of a recent study investigating the effect of the oxidative stress inducer elesclomol [108]. The authors reported that elesclomol is able to impair both GSC and GdEC survival in vitro. In a mouse model of GSC-derived brain xenograft, they showed that elesclomol inhibits tumor growth and enhances the antitumor effect of TMZ, interfering with GSC survival and motility signals [108].

Recently, it has been demonstrated that regorafenib, a multikinase inhibitor that targets angiogenic kinases (VEGFR 1–3 and PDGFR-b), as well as, oncogenic kinases (c-KIT, RET, FGFR, and Raf), has an impact on the endothelial transdifferentiation of GSCs [137,138]. Deshors et al. demonstrated that this drug significantly reduces tumor-derived ECs as well as their proangiogenic abilities, both in vitro and in vivo. Intriguingly, the authors also demonstrated that regorafenib can inhibit the irradiation-induced transdifferentiation of GSCs in vivo. Indeed, the mice irradiated and treated with regorafenib had significantly fewer functional blood vessels than the irradiated mice [138]. Regorafenib has recently been approved for second-line treatment of recurrent GBM. A phase II/III response adaptive randomization platform trial (NCT03970447, GBM AGILE) assessing the efficiency of regorafenib in newly diagnosed and recurrent GBM is ongoing to confirm the data that brought it to a preliminary authorization (https://clinicaltrials.gov/ct2/show/NCT03970447 (accessed on 31 August 2022)).

## 3. Conclusions

Due to its abundant vascularization, GBM is a candidate for anti-angiogenic therapy; however, despite the promising results in preclinical models and the observed early clinical response, i.e., the prolongation of PFS and a reduction in corticosteroid use, anti-angiogenic therapy failed to show a survival benefit in GBM patients whose tumor progresses. An improved understanding of the mechanisms of resistance to anti-angiogenic therapies and a better selection of patients will be crucial for physicians to identify effective strategies and improve outcomes for GBM patients.

## Data Availability

Not applicable.

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
