# Peer review of "Glioblastoma-Specific Strategies of Vascularization: Implications in Anti-Angiogenic Therapy Resistance"

_jpm, 2022, doi:10.3390/jpm12101625_

Round 1

Reviewer 1 Report

Title: Glioblastoma-specific strategies of vascularization: implications in anti- angiogenic therapy resistance

In this review, Mariachiara et al. well summarized the processes of neovascularization associated with glioblastoma, and to what extent this neovascularization contributes to therapy resistance and to limit the efficacy of current anti-angiogenic therapies in GBM. The authors mainly focused on three aspects, namely vessel co-option, vasculogenic mimicry and cell transdifferentiation. For each aspect, the authors first explained its involvement in therapy resistance  in GBM, followed with the potential strategies to overcome the resistance to anti-angiogenic treatment. It also included the information of current ongoing clinical trials either with individual new drug or the combination of new drug with conventional anti-angiogenic drugs for GBM treatment. Overall, this review is very informative and well-structured. I suggest to accept with current condition only with correcting some typo and grammar errors.

Author Response

Reviewer #1

Comments and Suggestions for Authors

In this review, Mariachiara et al. well summarized the processes of neovascularization associated with glioblastoma, and to what extent this neovascularization contributes to therapy resistance and to limit the efficacy of current anti-angiogenic therapies in GBM. The authors mainly focused on three aspects, namely vessel co-option, vasculogenic mimicry and cell transdifferentiation. For each aspect, the authors first explained its involvement in therapy resistance  in GBM, followed with the potential strategies to overcome the resistance to anti-angiogenic treatment. It also included the information of current ongoing clinical trials either with individual new drug or the combination of new drug with conventional anti-angiogenic drugs for GBM treatment. Overall, this review is very informative and well-structured. I suggest to accept with current condition only with correcting some typo and grammar errors.

We thank this reviewer for the positive comment. We have corrected typo and grammar errors as suggested.

Reviewer 2 Report

This article provides an excellent current view of tumor angiogenesis/vascularization in glioblastomas.  It summarizes the field's progress and provides a basis for future studies with clarity and clinical insight.

Minor suggestions for the authors to consider pertain only to text editing as follows: 3rd line on p 4 of 24, "...are associated with vessel co-option and..."; 5th line on p 7 of 24, "Over the last few years,..."; 6th line in 2.2. Vasculogenic mimicry's 1st paragraph on p 7 of 24, "Doppler untrasonography..."; 23rd line on p 8 of 24, "...(EphA2) are highly expressed in..."; 4th line in 2.2.2 Vasculogenic mimicry... on p 9 of 24, "...VM-related proteins associated with the signaling..."; 15th and 16th lines on p 10 of 24, "...form tube-like structures on extracellular matrix, as a model..."; 16th line on p 12 of 24, "...cells, whose frequency in histology..."; 20th line on p 12 of 24, "...it has been proven that hypoxia...".

Author Response

Reviewer #2

Comments and Suggestions for Authors

This article provides an excellent current view of tumor angiogenesis/vascularization in glioblastomas.  It summarizes the field's progress and provides a basis for future studies with clarity and clinical insight.

Minor suggestions for the authors to consider pertain only to text editing as follows: 3rd line on p 4 of 24, "...are associated with vessel co-option and..."; 5th line on p 7 of 24, "Over the last few years,..."; 6th line in 2.2. Vasculogenic mimicry's 1st paragraph on p 7 of 24, "Doppler untrasonography..."; 23rd line on p 8 of 24, "...(EphA2) are highly expressed in..."; 4th line in 2.2.2 Vasculogenic mimicry... on p 9 of 24, "...VM-related proteins associated with the signaling..."; 15th and 16th lines on p 10 of 24, "...form tube-like structures on extracellular matrix, as a model..."; 16th line on p 12 of 24, "...cells, whose frequency in histology..."; 20th line on p 12 of 24, "...it has been proven that hypoxia...".

We thank this reviewer for the enthusiastic comment. We have corrected typo and grammar errors as suggested.